

# Targeting squalene epoxidase in the treatment of metabolic-related diseases: current research and future directions

Mingzhu Chen[1], Yongqi Yang[2], Shiting Chen[1], Zhigang He[1] and Lian Du[1]

[1] School of Basic Medical Sciences, Chengdu University of Chinese Medicine, Chengdu, Sichuan Province, China
[2] Harbin Medical University, Department of Pharmacology, College of Pharmacy, Harbin, Heilongjiang Province, China

## ABSTRACT

Metabolic-related diseases are chronic diseases caused by multiple factors, such as genetics and the environment. These diseases are difficult to cure and seriously affect human health. Squalene epoxidase (SQLE), the second rate-limiting enzyme in cholesterol synthesis, plays an important role in cholesterol synthesis and alters the gut microbiota and tumor immunity. Research has shown that SQLE is expressed in many tissues and organs and is involved in the occurrence and development of various metabolic-related diseases, such as cancer, nonalcoholic fatty liver disease, diabetes mellitus, and obesity. SQLE inhibitors, such as terbinafine, NB598, natural compounds, and their derivatives, can effectively ameliorate fungal infections, nonalcoholic fatty liver disease, and cancer. In this review, we provide an overview of recent research progress on the role of SQLE in metabolic-related diseases. Further research on the regulation of SQLE expression is highly important for developing drugs for the treatment of metabolic-related diseases with good pharmacological activity.

## INTRODUCTION

Rapid economic growth, coupled with changes in people's dietary habits and lifestyles, has led to surges in mortality and morbidity associated with metabolism-related diseases, placing a substantial burden on public health systems and medical resources (*Saklayen, 2018*). The currently available medications for the prevention and treatment of these diseases require long-term use, and there are issues of general efficacy, drug resistance, and adverse side effects. Therefore, the exploration of innovative approaches to prevent and control these conditions is important. Studies have identified squalene epoxidase (SQLE) as a key regulator of various physiological processes, including cholesterol biosynthesis (*Parris et al., 2014*; *Padyana et al., 2019*), modulation of the gut microbiota (*Li et al., 2022a*), and the regulation of tumor immunity (*You et al., 2022*). Consequently, the involvement of SQLE has been implicated in the development of various conditions, such

Corresponding author
Lian Du, dulian@cdutcm.edu.cn

as fungal infections (*Ryder, 1992*), nonalcoholic fatty liver disease (NAFLD) (*Liu et al., 2021*), cancer (*Zhang et al., 2024*), diabetes mellitus (DM) (*Ge et al., 2020*), and obesity (*Ding et al., 2015*). Understanding the precise role of SQLE in the pathogenesis of these diseases and its potential as a therapeutic target is essential for devising effective preventive and therapeutic strategies. By elucidating the regulatory mechanisms and molecular pathways mediated by SQLE in metabolism-related diseases, we can establish a solid theoretical foundation for developing targeted interventions. Here, we aim to provide a comprehensive overview of SQLE, beginning with its structural characteristics and pivotal role in cholesterol synthesis. We discuss the regulation, functional importance, and clinical implications of SQLE in the context of metabolism-related disorders. Finally, we address the latest advancements in the use of SQLE as a targeted therapeutic agent, thus paving the way for future research directions and clinical applications.

### Intended audience and need for this review

SQLE is the second rate-limiting enzyme of cholesterol synthesis and plays an important role in cholesterol synthesis, alteration of the intestinal gut microbiota, and tumor immunity. In addition, SQLE is expressed in many tissues and organs and is involved in the occurrence and development of a variety of metabolic-related diseases. However, current studies on its function, expression, role in metabolic-related diseases and the development of clinical applications are not comprehensive. This article reviews advancements in research into the role of SQLE in metabolic-related diseases in recent years. An in-depth study of SQLE expression regulation is of particular importance for the treatment of metabolic-related diseases with good pharmacological activity.

## SURVEY METHODOLOGY

The authors conducted an in-depth search on PubMed, Web of Science, and the Foreign Medical Literature Retrieval Service. The search was carried out by combining subject words and free words, and the following heading terms were used when performing the search: "squalene epoxidase", "squalene epoxidase inhibitors", "cancer", "nonalcoholic fatty liver disease", "diabetes mellitus", "obesity", "cholesterol synthesis", "structure", "activity regulation", "function", "transcriptional regulation", "posttranscriptional regulation" and "metabolic disease". This review is based on published works, which have been classified, organized, and searched by title, abstract, and full text.

## STRUCTURE, ACTIVITY REGULATION, AND FUNCTION OF SQLE

### Structure of SQLE

SQLE was first discovered in rat liver microsomes in 1969 (*Yamamoto & Bloch, 1970*). Like most cholesterol enzymes, SQLE is located in the endoplasmic reticulum or on lipid droplets, and the *SQLE* gene is located on human chromosome 8q24.13 (*Nagai et al., 1997*).

   As an essential lipid component of mammalian cell membranes, cholesterol is critical for cell survival and proliferation and coordinates multiple membrane receptor signaling
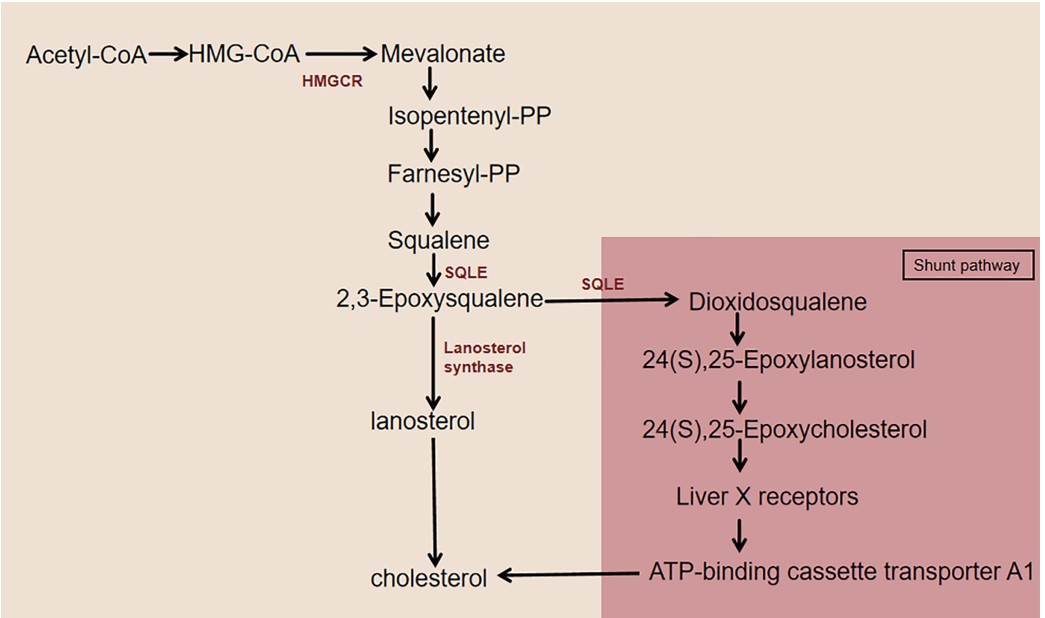

**Figure 1 Cholesterol synthesis pathway centered on SQLE.** This figure illustrates the simplified cholesterol biosynthesis pathway with a focus on SQLE. The pathway features key rate-limiting enzymes, including HMGCR and SQLE. HMGCR catalyzes the conversion of HMG-CoA to MVA. SQLE first converts squalene to 2,3-epoxysqualene. This intermediate then undergoes further enzymatic reactions to eventually produce cholesterol. When lanosterol synthase activity is low, SQLE diverts 2,3-epoxysqualene to dioxidosqualene, which is then converted to 24(S),25-epoxylanosterol and subsequently to 24(S),25-epoxycholesterol. 24(S),25-Epoxycholesterol binds to LXR, activating it and leading to the upregulation of ABCA1, which enhances cholesterol efflux.

pathways by maintaining the stability of lipid rafts (*Dang & Cyster, 2019*). Almost all mammals can synthesize cholesterol from acetyl coenzyme (acetyl-CoA) through a series of 20 enzymatic reactions, including the mevalonate (MVA) pathway, SQLE biosynthesis, and subsequent reactions (Fig. 1). 3-Hydroxy-3-methylglutaryl-CoA reductase (HMGCR) and SQLE are two key rate-limiting enzymes for cholesterol synthesis (*Howe et al., 2017*). The HMGCR-mediated mevalonate pathway is a key step in the *de novo* synthesis of cholesterol in the body. HMGCR-like SQLE activity is precisely regulated by intracellular cholesterol levels *via* feedback, which results in a second rate-limiting step in cholesterol synthesis (*Ryder, 1988*, *1991*). The two rate-limiting enzymes, HMGCR and SQLE, are important factors in the cholesterol synthesis pathway. SQLE is responsible for the first oxidative step in cholesterol synthesis: it oxidizes squalene to 2,3-epoxysqualene. When the activity of the enzyme lanosterol synthase, which converts 2,3-epoxysqualene to lanosterol, is low, SQLE also converts 2,3-epoxysqualene to dioxidosqualene. The end product of this shunt pathway, 24(S),25-epoxycholesterol, is a ligand for the liver X receptor (LXR), which increases ATP-binding cassette transporter protein A1 (ABCA1) levels to promote cholesterol efflux (*Seiki & Frishman, 2009*; *Gill et al., 2011*). Therefore, the catalytic reaction of SQLE is essential for cholesterol synthesis.

## Regulation and function of SQLE

SQLE is a direct target of sterol regulatory element-binding proteins (SREBPs). There are three SREBPs: SREBP1a, SREBP1c, and SREBP2, of which SREBP2 is a transcription factor that regulates genes involved in cholesterol synthesis and homeostasis in a cholesterol-dependent manner (*Bengoechea-Alonso, Aldaalis & Ericsson, 2022*). SQLE proteins contain cholesterol-sensing structural domains that regulate the proteasomal degradation of SQLE. SQLE expression in cells is subject to complex regulatory systems, including transcription, posttranscriptional regulation at the mRNA level, and posttranslational regulation.

## Transcriptional regulation

The transcription factor SREBP2 directly regulates the mRNA levels of enzymes involved in cholesterol metabolism, such as SQLE, by binding to sterol regulatory element (SRE) sequences in the promoters of target genes (*Brown, Radhakrishnan & Goldstein, 2018*). When the cholesterol level in the endoplasmic reticulum (ER) is less than 5% of the total intracellular lipid level, SREBPs are activated, and SREBP cleavage-activating protein (SCAP) undergoes a conformational change to dissociate from the Insig-1 protein. The SCAP–SREBP2 complex is then disassembled and detached from the ER membrane and transported to the Golgi complex *via* coat protein complex II (COPII) vesicles. This protein is subsequently cleaved by site-1 protease (S1P) and site-2 protease (S2P) protein hydrolase cleavage and converted to the activated nuclear form SREBP2 (nSREBP2). Immediately thereafter, nSREBP2 enters the nucleus as a homodimer and binds to SRE in the promoter of the target gene *SQLE* to increase SQLE mRNA levels (*Griffiths & Wang, 2021*). In addition, oxygen sterol-binding protein-like 2 (OSBPL2) deficiency promotes nuclear entry of the specificity protein 1 (SP1) transcription factor and SREBP2 in the SQLE promoter to upregulate SQLE expression and increase cholesterol and cholesteryl ester accumulation through inhibition of the AMP-activated protein kinase (AMPK) pathway (*Zhang et al., 2019*). In addition, the transcription factors nuclear factor Y (NF-Y) and SP1 act synergistically with nSREBP2 to upregulate *SREBP2* gene expression.

Oncogenes/tumor-suppressor genes are also involved in SQLE transcriptional regulation independent of SREBP2. The proto-oncogene *MYC* upregulates SQLE transcription by binding to the response element 1 (RE1) of the SQLE gene in cancer (*Yang et al., 2021*). The *p53* gene is a common antioncogene in human cancers with a high ability to control cholesterol synthesis. Under low sterol conditions, p53 mediates SREBP2 regulation of SQLE transcription. In castration-resistant prostate cancer (CRPC), PTEN/ p53-deficient tumors are dependent on cholesterol metabolism and upregulate SQLE by activating SREBP2 transcription to satisfy the cholesterol demand of tumor cells and promote the growth and progression of CRPC (*Shangguan et al., 2022*). Under normal sterol conditions, p53 directly represses SQLE expression in an SREBP2-independent manner, and p53 represses SQLE transcription through direct binding to Response Element 2 (RE2) in hepatocellular carcinoma (HCC) cells. p53 deficiency increases SQLE expression even when ER membrane cholesterol levels are normal or elevated (*Sun et al., 2021*).

## Posttranscriptional regulation

A large body of evidence suggests that the expression profiles of long noncoding RNAs (lncRNAs), microRNAs (miRNAs), and circular RNAs (circRNAs) are commonly dysregulated in human cancers. miRNA-133b (*Qin et al., 2017*; *Wang et al., 2022*), miRNA-205 (*Kalogirou et al., 2021*), miRNA-612 (*Liu et al., 2020*), miRNA-579-3p (*Qian et al., 2023*), miRNA-1179 (*Li, Jiang & Zhao, 2023*), miRNA-584-5p (*Li et al., 2022b*) and miRNA-363-3p (*You et al., 2022*) can interact with SQLE mRNAs, act as negative regulators of gene expression at the posttranscriptional level, and play important roles in tumor cell differentiation, proliferation, and apoptosis through the miRNA/SQLE axis. Some lncRNAs and circRNAs act as competing endogenous RNAs (ceRNAs) that sponge their corresponding miRNAs, thus decreasing their inhibitory effects on their targets (*Tay, Rinn & Pandolfi, 2014*; *Qian et al., 2023*). *Qin et al. (2021)* reported that in breast cancer, the lncRNA lnc030 is highly expressed in cancer stem cells and is positively correlated with SQLE expression. Poly(rC)-binding protein 2 (PCBP2) consists of 3 K homology (KH) domains (*Silvera, Gamarnik & Andino, 1999*). Lnc030 interacts with the KH2 of PCBP2, and the 3′ untranslated region (3′UTR) of SQLE mRNA binds to the KH3 of PCBP2. Lnc030 synergistically enhances the stability of SQLE mRNA with PCBP2. This leads to an increase in cholesterol synthesis, which, in turn, activates the PI3K/AKT signaling pathway and is involved in the regulation of the stemness characteristics of breast cancer stem cells (BCSCs). miRNA-133b and miRNA-205 were reported to reduce SQLE mRNA levels more rapidly by binding to the 3′UTR of SQLE mRNA (*Qin et al., 2017*; *Kalogirou et al., 2021*; *Wang et al., 2022*). circ_0000182 was shown to cause SQLE overexpression by sponging miRNA-579-3p. This miRNA then loses the ability to regulate SQLE and thus promotes cholesterol synthesis and proliferation in gastric adenocarcinoma cells (*Qian et al., 2023*). In summary, miRNAs, lncRNAs, and circRNAs can exert their effects on various cancers by modulating the expression of SQLE (Table 1). Targeting the interactions between miRNAs, lncRNAs, circRNAs, and SQLE holds promise for providing new strategies for the clinical treatment of patients with cancer.

## Posttranslational regulation

SQLE is also posttranslationally regulated. The regulation of SQLE protein stability and activity, mainly through the cholesterol membrane-associated ring-CH-type finger 6 (MARCH6)-proteasomal degradation axis, is dependent on specific structural domains in SQLE proteins (*Sharpe, Coates & Brown, 2020*). These domains include the cholesterol-dependent amphiphilic helical structure formed by the Gln62-Leu73 sequence (*Chua et al., 2017*), which can be embedded in the hydrophobic interior of membranes while interacting with the hydrophilic environment at the membrane surface. It regulates SQLE by interacting with cholesterol molecules and plays an important role in cholesterol-dependent degradation. The first 100 amino acids of SQLE (SQLE N-100) are attached to the ER membrane in the form of a re-entrant loop (*Howe et al., 2015*), which senses cholesterol in the cytoplasm. When intracellular cholesterol accumulates, the anchoring of the SQLE protein to the ER membrane tightens. This results in partial exposure of the amphipathic helical structure of the Gln62-Leu73 sequence to the

**Table 1 Interactions of miRNAs, lncRNAs, and circRNAs with SQLE and their effects on cancer cells.** miRNAs, lncRNAs, and circRNAs can all exert their effects on various cancers by influencing the expression of SQLE. SQLE is highly expressed in multiple types of cancer cells and promotes tumor progression. miRNA-205, miRNA-133b, miRNA-579-3p, miRNA-584-5p, miRNA-1179, miRNA-363-3p, and miRNA-612 are down-regulated in cancers and negatively correlated with SQLE expression. Conversely, lncRNA 030 and circRNA are downregulated in cancers and are positively correlated with SQLE expression.

| RNA type | RNA identifier | Cancer cells | miRNA expression level SQLE expression level | Function | Overexpressed | Ref |
|---|---|---|---|---|---|---|
| miRNA | miRNA-205 | Prostate cancer | Downregulated | Overexpressed | Promotion of cell proliferation and androgen receptor | Kalogirou et al. (2021) |
| miRNA | miRNA-133b | Pancreatic cancer | Downregulated | Overexpressed | Promotion of cell proliferation, migration, and invasion | Wang et al. (2022) |
| miRNA | miRNA-133b | Esophageal squamous cell carcinoma | Downregulated | Overexpressed | Promotion of cell proliferation, migration, and invasion | Qin et al. (2017) |
| miRNA | miRNA-579-3p | Gastric adenocarcinoma | Downregulated | Overexpressed | Promotion of cell proliferation | Qian et al. (2023) |
| miRNA | miRNA-584-5p | Head and neck squamous cell carcinomas | Downregulated | Overexpressed | Promotion of cell proliferation, migration, and invasion | Li et al. (2022b) |
| miRNA | miRNA-1179 | Nasopharyngeal carcinoma | Downregulated | Overexpressed | Promotion of cell proliferation and inhibition of apoptosis | Li, Jiang & Zhao (2023) |
| miRNA | miRNA-363-3p | Pancreatic Cancer | Downregulated | Overexpressed | Promotion of cell proliferation, regulation of tumor immune cell infiltration and expression of immune checkpoints | You et al. (2022) |
| miRNA | miRNA-612 | Hepatocellular carcinoma | Downregulated | Overexpressed | Promotion of cell invadopodia, epithelial-mesenchymal transition, migration, and invasion | Liu et al. (2020) |
| lncRNA | lnc RNA 030 | Breast cancer | Upregulated | Overexpressed | Maintains BCSCs Stemness | Qin et al. (2021) |
| circRNA | circ_0000182 | Gastric adenocarcinoma | Upregulated | Overexpressed | Promotion of cell proliferation | Qian et al. (2023) |

cytoplasmic environment, preventing proteasomal degradation. The ubiquitination process also requires the ubiquitin-conjugating enzyme E2 J2 (UBE2J2). MARCH6 approaches the SQLE protein and recognizes serine residues near the cholesterol-dependent amphiphilic helix in the SQLE protein. UBE2J2 works in concert with MARCH6 to attach the ubiquitin molecule from the E1 enzyme to the SQLE protein to be ubiquitinated to label it as a protein to be degraded so that it becomes a ubiquitin-protease substrate of the degradation system. The valosin-containing protein (VCP) is then recruited to extract the ubiquitinated SQLE substrate, where it dissociates from the ER. VCP then cooperates with other proteins to mediate entry into the proteasomal degradation pathway. Excess cholesterol can stimulate SQLE degradation by inhibiting MARCH6 self-degradation (*Sharpe et al., 2019*).

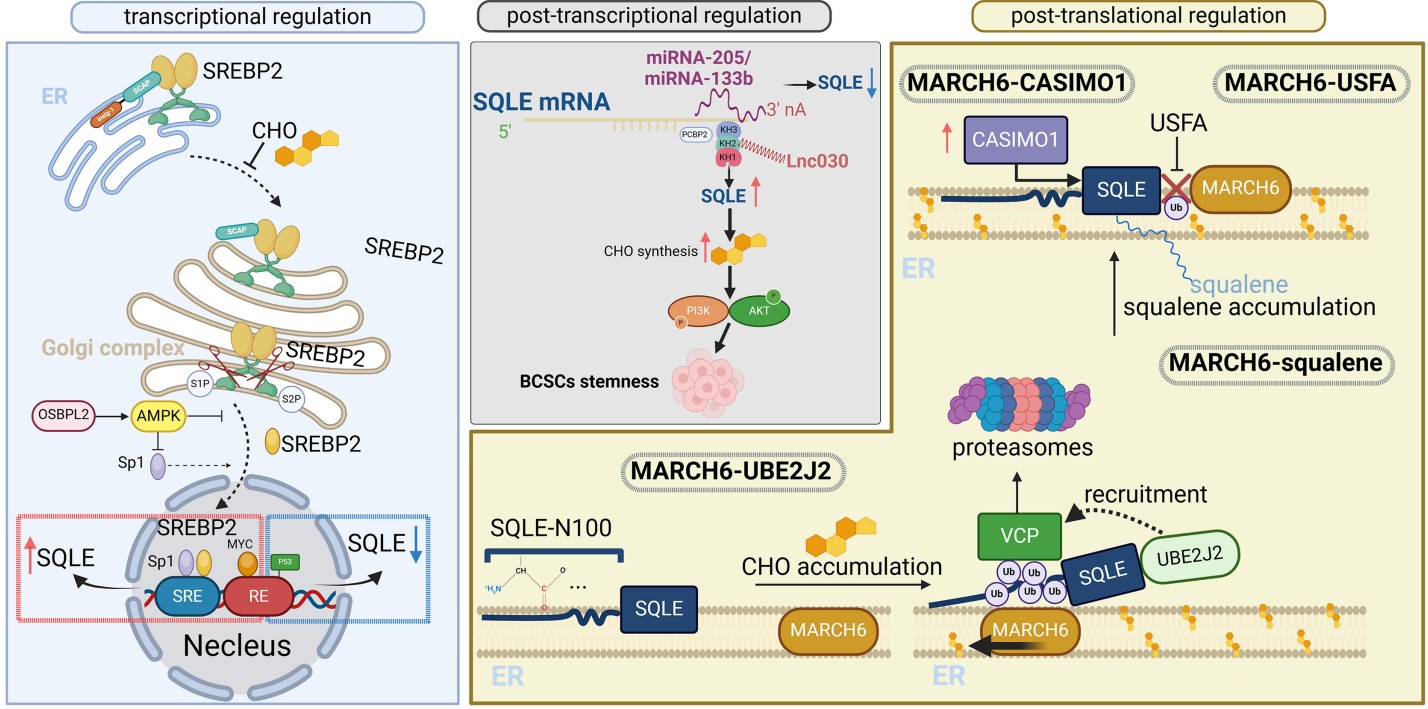

**Figure 2 Transcriptional, posttranscriptional, and posttranslational regulation of SQLE.** SQLE can be regulated at the transcriptional level by SREBP2, which is a transcription factor involved in cholesterol synthesis and homeostasis. At the posttranscriptional level, gene expression is regulated through interactions between miRNAs/lncRNAs and SQLE mRNAs. At the posttranslational level, SQLE is regulated through cholesterol-dependent feedback mechanisms. The figure is adapted from *Zou et al. (2022)* with modifications. Created in BioRender.

The proteasomal degradation pathway of MARCH6-VCP can be regulated independently of cholesterol. The N-terminus of SQLE is partially degraded through a unique ubiquitination pathway, which leads to the conversion of full-length SQLE to a truncated form. The enzyme activity of the truncated SQLE is associated with cholesterol resistance, implying that the function of SQLE is not completely lost under high-cholesterol conditions. Thus, the function of this gene may complement cholesterol metabolism under cancer conditions (*Coates, Capell-Hattam & Brown, 2021*). In addition to cholesterol, squalene directly binds to the SQLE N-100 structural domain to change its conformation. This results in the inability of MARCH6 to ubiquitinate SQLE or label it for degradation, thus increasing the stability of SQLE (*Nathan, 2020*). Unsaturated fatty acids (USFAs) can also stabilize SQLE levels *via* the regulatory blockade of SQLE ubiquitination by MARCH6. Upregulation of the cancer-associated microprotein CASIMO1 increases SQLE levels by interacting with SQLE proteins (*Polycarpou-Schwarz et al., 2018*).

To better understand the regulatory mechanisms of SQLE, its regulation at the transcriptional, posttranscriptional, and posttranslational levels is illustrated in Fig. 2 (adapted from *Zou et al., 2022*). Overall, the main mechanism of SQLE regulation is the cholesterol-dependent feedback regulation of SQLE by the SREBP2 transcriptional and ubiquitin proteasomal degradation pathways. SREBP2 activation in tumor tissues leads to high SQLE expression. The activation of oncogenes, deletion of cancer suppressor genes,

and deletion of miRNAs and cancer-associated proteins directly or indirectly upregulate SQLE expression. Different tumor cells or specific subpopulations have specific SQLE regulatory mechanisms (*Du, Rokavec & Hermeking, 2023*).

Because of the crucial role of the 2,3-epoxysqualene derivative lanosterol in fungal membrane synthesis, SQLE has long been investigated as an antifungal target with increasing relevance to human health and disease (*Astruc et al., 1977*). Many studies have shown that SQLE can meet the high energy requirements for the rapid growth of diseased cells, and the dysregulation of SQLE has been found in a variety of metabolic-related diseases. This has become a hot topic in the field of targeted diagnosis and therapy (*Ryder, 1992*).

## The role and clinical significance of SQLE in metabolism-related diseases

### NAFLD

In recent years, NAFLD has replaced viral hepatitis as the most common chronic liver disease in China, and the number of patients with NAFLD in China is expected to reach 314 million by 2030 (*Nan et al., 2021*). NAFLD is the hepatic manifestation of metabolic syndrome and includes simple steatosis to nonalcoholic steatohepatitis (NASH).

*Liu et al. (2021)* reported that SQLE was closely related to the development of NASH and that SQLE expression levels were significantly increased in NAFLD model mice. Hepatocyte-specific overexpression of SQLE triggers spontaneous insulin resistance and NAFLD and induces the activation of SREBP1c, acetyl-CoA carboxylase (ACC), fatty acid synthase (FASN), and stearoyl coenzyme A (SCA) through the promotion of cholesterol synthesis and accumulation, and the binding of carbonic anhydrase 3 (CA3) and stearoyl-CoA desaturase1 (SCD1) leads to the expression of lipogenesis and triglyceride biosynthesis genes, thereby inducing *de novo* hepatic lipogenesis and the activation of the NF-κB inflammatory pathway, which drives the pathogenesis and progression of NASH. Therefore, SQLE and CA3 can be used as nominal markers for the diagnosis of NAFLD or NASH. Studies have shown that SQLE is significantly upregulated in patients with NAFLD-related HCC; mouse hepatocyte-specific overexpression of SQLE drives cholesterol biosynthesis and the NADP/NADPH ratio, triggering an oxidative stress response that activates the DNA methyltransferase 3A-mediated PTEN/PI3K/AKT/mTOR signaling pathway, driving NAFLD-related HCC carcinogenesis; and the SQLE inhibitor terbinafine inhibits NAFLD-related HCC cell proliferation and tumor development in a mouse model (*Liu et al., 2018*). In addition, *Sun et al. (2021)* reported that p53 regulates cholesterol synthesis by inhibiting the transcription of SQLE, thus exerting an inhibitory effect on NAFLD-related HCC.

Through the above studies, we revealed the mechanism of the development of NASH and NAFLD-related HCC and confirmed the importance of SQLE as a key driver and new drug target. However, NAFLD is not a simple benign disease. According to a global data assessment in 2015, the number of deaths of patients with NAFLD-related HCC in China accounted for 10.5% of the total number of HCC deaths in the same period (*Nan et al., 2021*). Consistent with the NAFLD mouse model, SQLE overexpression also

activated cholesterol and lipid biosynthesis in the human NAFLD cohort through the upregulation of hepatic sterol regulatory element-binding protein 1 (SREBF1), FASN, and SCD1 expression to drive the progression of NASH (*Liu et al., 2021*). Moreover, the application of the SQLE inhibitor terbinafine eliminated the accumulation of hepatic cholesterol, triglycerides, and free fatty acids in SQLE-overexpressing mice, normalized insulin sensitivity, and improved insulin tolerance test (ITT) and serum insulin levels. The combination of the CA3 inhibitor acetazolamide for the treatment of NASH improved the effectiveness of both drugs due to their synergistic effects, further confirming that SQLE/CA3 is a new target for the diagnosis and treatment of NASH.

### Cancer

There is growing evidence that cancer is also a metabolic disease involving tumor cell proliferation, energy metabolism, and dysregulation of immune surveillance. Reprogramming of cholesterol metabolism in tumor cells involves synthesis, uptake, esterification, efflux and transformation processes, which promote tumorigenesis (*Huang, Song & Xu, 2020*). Studies have shown that SQLE is aberrantly expressed in a variety of malignant tumors and influences tumor cell proliferation, migration or invasion through pathways such as cholesterol synthesis, tumor immune infiltration and immunotherapy, and intestinal ecology. The level of SQLE expression may be correlated with aspects of cancerous tissue lesions, ethnicity, and disease stage (*D'Arcy et al., 2015*; *Jun et al., 2021*; *He et al., 2021*; *Li et al., 2022a*). The expression of SQLE is significantly upregulated in nasopharyngeal carcinomas (*Li, Jiang & Zhao, 2023*), leukemias (*Song et al., 2022*), pancreatic cancer (*Wang et al., 2022*), and hepatocellular carcinoma (*Sui et al., 2015*).

As a key enzyme in cholesterol synthesis, SQLE activity determines the abundance of cholesterol and cholesterol derivatives, which promotes tumor growth through the accumulation of cholesterol/cholesterol esters and, subsequently, multiple oncogenic pathways. For example, p53 directly reduces SQLE expression in an SREBP2-independent manner, inhibiting cholesterol production *in vivo* and *in vitro* and leading to tumor growth inhibition (*Sun et al., 2021*). In prostate cancer cells, the involvement of SQLE in cholesterol synthesis consumes large amounts of NADPH and activates DNA methyltransferase 3A (DNMT3A). This triggers loss of function or reduced expression of the *PTEN* gene, which drives cholesteryl ester accumulation and subsequent sterol O-acyltransferase 1 (SOAT1) activation *via* the PTEN/PI3K/AKT/mTOR pathway (*Yue et al., 2014*). These intertwined cascade reactions amplify the oncogenic effects of SQLE. Aberrant activation of the ERK signaling pathway promotes cancer cell growth and proliferation, apoptosis, invasion and metastasis, and angiogenesis and is tightly associated with cancer development. The control of cholesterol biosynthesis associated with SQLE is markedly increased in patients with colorectal cancer (CRC). *He et al. (2021)* revealed that SQLE deficiency in CRC reduces intracellular cholesterol levels and decreases osteotriol (the active form of vitamin D3), leading to reduced levels of cytochrome P450 family 24 subfamily a member 1 (CYP24A1), the inhibition of ERK phosphorylation and CRC cell proliferation.

In addition to the cell-intrinsic effects of SQLE, it may also play a role in tumor growth through host–microbiota interactions. The intestinal flora is a diverse and surprisingly numerous microbial community present in the human gut that is associated with inflammatory/immune diseases, metabolic disorders, and malignancies (*Toya et al., 2021*). In recent years, SQLE has been shown to be involved in tumor growth regulation by affecting the composition and function of gut microbes, thereby influencing metabolite production and modulating the immune response. *Li et al. (2022a)* reported that SQLE-induced dysregulation of gut microbes promotes intestinal barrier dysfunction and proliferation of the colonic epithelium in germ-free mice and that the metabolism of secondary bile acids disrupts intestinal barrier function. Additionally, these authors reported that the downregulation of the tight junction proteins Jam-c and occludin causes a "leaky gut", which ultimately induces a proinflammatory response, and that the transplantation of feces from SQLE transgenic mice into germ-free mice impairs the intestinal function and proliferation of the epithelium.

SQLE plays an important role in tumor immunomodulation, and an algorithmic analysis of databases revealed that SQLE mRNA was expressed at higher levels in head and neck squamous cell carcinoma (HNSCC) tissues than in normal tissues (*Liu, Fang & Liu, 2021*). The expression of SQLE in patients with glioblastoma (GBM) was significantly correlated with tumor-infiltrating lymphocytes, immune stimulants, immunosuppressants, and MHC molecules (*Ye et al., 2023*). Experimental findings demonstrated that SQLE expression was upregulated in patients with pancreatic adenocarcinoma (PAAD) and was negatively correlated with prognosis (*You et al., 2022*). SQLE can affect the immune microenvironment and immunotherapy outcome of patients with PAAD by regulating the infiltration of tumor immune cells and the expression of immune checkpoint therapy (ICT), and this type of metabolic intervention-based immunotherapy is beneficial for overcoming the bottleneck of cancer treatment.

SQLE significantly influences the development and progression of several cancers by modulating cholesterol synthesis, the gut microbiota, and the immune microenvironment. These findings provide new perspectives for clinical practice and highlight the potential value of SQLE in different cancer types. SQLE is involved in hormone signaling, and in prostate cancer, it is closely associated with high Gleason scores, correlates with metastasis, distinguishes tumors at high risk of metastasis, and is a strong predictor of fatal prostate cancer (*Stopsack et al., 2016, 2017*). In breast cancer, SQLE overexpression is usually associated with tumor aggressiveness, recurrence, and overall survival time, and breast cancers with amplification of 8q24.11–13 (a region that includes the *SQLE* gene) imply a poorer prognosis (*Parris et al., 2014*). The mRNA expression of SQLE has also been associated with a poorer prognosis in patients with estrogen receptor-positive (ER+) phase I/II breast cancer (*Helms et al., 2008*).

Poor drug response to letrozole and poor progression-free survival with adjuvant tamoxifen have been reported in patients with SQLE-overexpressing breast cancer (*Simigdala et al., 2016*). In HCC, the SQLE is an independent risk factor for overall survival, and high levels of SQLE expression significantly correlate with advanced tumor histological grade and elevated levels of alpha-fetoprotein (*Huang, Song & Xu, 2020*; *Shen,*

*Lu & Zhang, 2020*). Thus, the SQLE may serve as a novel prognostic biomarker. However, in patients with CRC, the prognostic value of SQLE is related to tumor progression. Higher levels of SQLE in tumors are associated with poorer overall survival in patients with stages II and III disease, but lower levels of SQLE expression in tumors with stage T4 or IV disease predict a poorer prognosis (*Kim, Kim & Kang, 2019*). In pancreatic cancer, high expression levels of SQLE and other genes involved in cholesterol production are associated with resistance to radiotherapy (*Souchek et al., 2014*). For squamous cell carcinoma of the lung, SQLE is closely associated with poor differentiation, clinical stage, and lymphatic metastasis, which predict a poor prognosis; thus, it has become a novel molecular marker for lung cancer (*Zhang et al., 2014*). In uveal melanoma and head and neck squamous cell carcinoma, SQLE was associated with poor prognosis (*Toya et al., 2021*; *Xu et al., 2019*). Daunorubicin-resistant leukemia cells express higher levels of SQLE than do daunorubicin-sensitive leukemia cells (*Stäubert et al., 2016*). In addition, cholesterol is a sex hormone precursor; therefore, SQLE overexpression is associated with adverse effects of hormone therapy. In conclusion, high levels of SQLE in most tumors predict poor prognosis, including tumor recurrence, tumor metastasis, and a short overall survival time (*Kim, Kim & Kang, 2019*; *Jiang et al., 2022*). The role of SQLE in tumor development and progression has been demonstrated through basic research and clinical analyses, and SQLE may be a new target for cancer therapy.

### DM

DM is a complex, chronic metabolic disease characterized by persistent hyperglycemia and widespread disturbances in glucose, protein, and lipid metabolism (*Singh et al., 2021*). In recent years, transcriptomics studies have provided a novel perspective for elucidating the molecular pathogenesis of DM. In particular, the key role of the *SQLE* gene in the pathogenesis of DM has attracted widespread attention.

In a mouse model of diabetes, analysis of differentially expressed genes (DEGs) in liver tissue revealed 27 significantly altered genes, with a particularly marked upregulation of *SQLE* gene (*Ge et al., 2018*). This finding aligns with the later studies, which confirmed the involvement of SQLE as a core differential protein in the pathogenesis of DM. Notably, whereas the RNA-seq and qRT-PCR results from the liver revealed the downregulation of SQLE mRNA levels, Western blot analysis revealed significant upregulation of SQLE protein expression. This apparent contradiction suggests that there may be a complex posttranscriptional regulatory mechanism involved in SQLE expression, such as increased protein stability or increased translational efficiency, leading to a sustained increase in its protein level (*Ge et al., 2020*). As a rate-limiting enzyme in the cholesterol biosynthesis pathway, SQLE is a key node in the regulation of cholesterol synthesis. In the diabetic state, SQLE overexpression leads to an abnormal increase in cholesterol synthesis, promotes cholesterol deposition in vessel walls, and dramatically increases the risk of atherosclerosis. These alterations not only exacerbate the pathologic process of DM but also may lead to serious cardiovascular complications (*He et al., 2020*). Therefore, the precise regulation of the SQLE is highly important for the balance of cholesterol metabolism in patients with diabetes and is expected to be a new target for diabetes treatment, especially in the

intervention of disorders of glucose and lipid metabolism and the prevention of related complications, which has potential clinical application value.

From an epidemiological perspective, the latest data from the International Diabetes Federation (IDF) highlight the global public health challenge of DM: the number of people with DM in the world will reach 537 million in 2021 and is projected to increase to 643 million by 2030, reaching 783 million by 2045 (*Einarson et al., 2018*). This alarming trend highlights the urgency of delving into the mechanisms of diabetes pathogenesis and developing novel therapeutic strategies. Further transcriptomic studies have revealed the crosscutting role of SQLE in the pathological processes of diabetes, obesity, and cardiovascular disease. The upregulation of SQLE expression in peripheral monocytes from patients with atherosclerosis reveals molecular links between obesity, chronic inflammation, type 2 diabetes mellitus (T2DM), and cardiovascular disease (*Ding et al., 2015*). Interestingly, weight loss induces remodeling of cholesterol metabolism pathways in monocytes, including the downregulation of SQLE expression, suggesting that SQLE may be a potential therapeutic target for obesity-associated T2DM. The Multi-Ethnic Study of Atherosclerosis (MESA) cohort study further confirmed that the expression pattern of the cholesterol metabolism gene network in circulating monocytes is a remarkable feature of obesity and chronic inflammatory states. This gene network is strongly associated with obesity and is also associated with T2DM and coronary artery calcification (CAC). In this network, SQLE is significantly upregulated, suggesting that it may influence the pathological associations among obesity, T2DM and CAC by regulating lipid metabolism (*Ding et al., 2015*).

In terms of therapeutic strategies, the SQLE inhibitor terbinafine has promising therapeutic potential. In a mouse model of SQLE overexpression, terbinafine markedly improved insulin sensitivity, optimizing the ITT results and serum insulin levels (*Liu et al., 2021*). More importantly, treatment with terbinafine in combination with acetazolamide effectively inhibited high-fat high-cholesterol (HFHC) or methionine-and choline-deficient (MCD) diet-induced lipid accumulation, including accumulations in triglycerides, free fatty acids, and serum triglycerides, resulting in marked improvements in insulin tolerance and glucose tolerance.

In summary, SQLE plays a pivotal role in the pathogenesis of DM and NASH. Therapeutic strategies targeting SQLE, especially SQLE inhibitors such as terbinafine, show promising therapeutic potential. These findings open new research directions for the precision treatment of DM, obesity and their related metabolic disorders and are expected to play an influential role in future clinical practice. However, more prospective randomized controlled trials are still needed to validate the long-term efficacy and safety from basic research to clinical application, as well as to explore the specific effects of SQLE in different tissues and cell types in depth to optimize targeted therapeutic strategies.

### Obesity

Obesity is a multifactorial chronic metabolic disease characterized by excessive accumulation and abnormal distribution of white adipose tissue (*Calcaterra et al., 2021*). As a stand-alone disease entity, obesity is also a major risk factor for many chronic

diseases, such as T2DM and cardiovascular diseases (*La Sala & Pontiroli, 2020*). The exponential growth of obesity-related healthcare costs globally, with the latest meta-analysis showing that their management accounts for 3% to 21% of national health budgets, underscores the urgency of developing effective prevention and treatment strategies (*Ahmed & Konje, 2023*).

In a molecular genetics study, the Fob3b obesity quantitative trait locus (QTL) was identified as one of the key regions affecting obesity-related phenotypes in mice. Among these contributors, the *SQLE* gene has attracted attention for its key role in cholesterol biosynthesis. Functional genomics studies revealed significant differences in SQLE expression levels between high-fat diet (HFD)-induced obese mice and homozygous recombinant mice harboring the Fob3b QTL region in low-fat diet-fed mice, suggesting an important role in the regulation of energy homeostasis and the maintenance of metabolic homeostasis. In-depth studies suggest that Fob3b allele variants may activate cholesterol biosynthesis by increasing SQLE expression or its enzymatic activity, leading to increased cholesterol deposition in adipocytes, promoting cellular hypertrophy and proliferation, and exacerbating obesity (*Stylianou et al., 2005*). Translational medicine research further supports the importance of SQLE in the pathogenesis of human obesity. Multiomics analysis revealed significant upregulation of SQLE expression in the adipose tissue of patients with obesity. In a 5-month weight loss intervention trial, participants lost an average of 6.7 ± 1.1% of their body weight and experienced a 33 ± 8% decrease in the homeostasis model assessment of insulin resistance (HOMA-IR) index, along with a marked decrease in SQLE expression levels (*Ding et al., 2015*). These findings reveal that an imbalance in cholesterol homeostasis in the obese state may be an important mechanism for metabolic disorders and that weight loss rebalances cholesterol metabolism gene network expression.

In exploring natural product strategies for the treatment of obesity, extracts of *Polygala tenuifolia* have received attention for their potential SQLE-modulating effects. *In vitro* studies have shown that some of the active components of these extracts directly inhibit SQLE activities. In the HFD-induced obese mouse model, a *Polygala tenuifolia* extract showed a marked antiobesity effect by regulating SQLE-mediated cholesterol biosynthesis, significantly lowering the cholesterol content in adipose tissue and inhibiting adipocyte differentiation and hypertrophy (*Wang et al., 2017*). These findings offer prospects for the development of novel natural product-based obesity treatment strategies, but in-depth studies are still needed to elucidate the underlying molecular mechanisms and assess long-term safety and clinical efficacy. Future research directions should include identifying the active components in the extracts of *Polygala tenuifolia* that specifically target SQLE, validating the metabolic regulation of SQLE in different tissues using conditional knockout models, and designing large-scale clinical trials to evaluate the effects of intervention strategies targeting SQLE.

## SQLE-targeted therapeutic strategies

SQLE has been reported to act as an oncogene in various cancers. Moreover, the dysregulation of SQLE has been associated with the inhibition of apoptosis and increased

**Table 2 Characteristics of different SQLE inhibitors.** This table summarizes the characteristics of various SQLE inhibitors, categorized by category, and includes their names, advantages, and limitations. The advantages and limitations of each inhibitor provide a detailed comparison of their usage and effectiveness. References are provided to support the data sources.

| Category | Name | Advantages | Limitations | Ref |
|---|---|---|---|---|
| Allylamine inhibitor | Terbinafine | Wide clinical application of antifungal drugs | Neurotoxicity, skin toxicity, poor resistance | *Padyana et al. (2019)*, *Brown, Chua & Yan (2019)*, *Nagaraja et al. (2020)* |
| Allylamine inhibitor | NB-598 | Highly specific to mammalian SQLE, high potency, time-dependent inhibition, competitive inhibition | Neurotoxicity, gastrointestinal and skin toxicity | *Padyana et al. (2019)*, *Nagaraja et al. (2020)* |
| Allylamine inhibitor | Cmpd-4″ | High potency, time-dependent inhibition, competitive inhibition | Gastrointestinal and skin toxicity | *Padyana et al. (2019)*, *Nagaraja et al. (2020)* |
| Allylamine inhibitor | FR194738 | Lipophilicity, favorable pharmacokinetics | Lack of clinical trials, safety and efficacy unknown | *Sawada, Washizuka & Okumura (2004)*, *Zhang et al. (2024)* |
| Natural Compounds and Derivatives | EGCG | Few side effects | Low bioavailability, short half-life | *Abe et al. (2000)* |
| Natural Compounds and Derivatives | Garlic and its derivatives | Antimicrobial and antioxidant properties | Slow, irreversible, low specificity | *Wagner, Toews & Morell (1995)*, *Laden & Porter (2001)*, *Gupta & Porter (2001)* |

cell proliferation and invasiveness, and a high abundance of SQLE in tumors indicates a poorer prognosis. As a novel and attractive therapeutic target for anticancer treatment, SQLE has been increasingly used in preclinical studies to reveal its antitumor effects and related mechanisms. The first SQLE-targeted inhibitors disrupted the synthesis of ergosterol in antifungal bodies, thereby killing or inhibiting the fungus (*Barrett-Bee & Dixon, 1995*). Currently, SQLE inhibitors are mainly classified as allylamines, natural compounds, or their derivatives (*Abe et al., 2000*; *Zhang et al., 2024*). Research on the novel use of established SQLE inhibitors is increasing, and targeting SQLE is considered a new and promising therapy for metabolic diseases (Table 2).

## Allylamine

Since the discovery in 1981 that naftifine has high broad-spectrum antifungal activity, it has become the cornerstone for the commercialization of next-generation inhibitor drugs, such as butenafine and tolnaftate (*Petranyi, Georgopoulos & Mieth, 1981*). Terbinafine is a common antifungal agent that has been proposed as a new therapeutic strategy for human cancers (*Chua, Coates & Brown, 2018*). The main SQLE inhibitor used in preclinical antitumor studies is terbinafine, which inhibits cell proliferation, induces G0/G1 cell cycle arrest, apoptosis, and autophagy by inhibiting SQLE or SQLE-independent inhibition, and slows tumor growth *in vivo* in a dose-dependent manner (*Xu et al., 2023*). In addition to the presence of neurological and dermal toxicity, poor drug resistance were detected for SQLE compared with NB-598 or Cmpd-4″, which had IC50 values in the range of 10–60 nM (*Padyana et al., 2019*; *Brown, Chua & Yan, 2019*; *Nagaraja et al., 2020*). The IC50 was determined to be 7.7 µM, with a maximum inhibition of 65% at an inhibitor

concentration of 100 μM, suggesting that terbinafine is not the best SQLE inhibitor (*Padyana et al., 2019*).

The compound NB598, obtained by modifying the aromatic moiety of terbinafine, is another highly specific inhibitor of mammalian SQLE, with the best responses in neuroblastoma and lung cancer and good drug sensitivity and high efficacy in small-cell lung cancer cell lines (*Mahoney et al., 2019*; *Padyana et al., 2019*). Further modification of NB598 yielded silyl derivatives that are also able to inhibit the enzymatic activity of SQLE. However, preclinical studies revealed that gastrointestinal and dermal toxicities were not tolerated by dogs and monkeys treated with a gavage of allylamine inhibitors (NB-598 and Cmpd-4″) for small cell lung cancer treatment (*Nagaraja et al., 2020*). Cmpd-4″ has been reported to share the same high potency as NB-598 for the time-dependent inhibition of SQLE but suffers from the same gastrointestinal and dermal toxicities (*Padyana et al., 2019*). Naftifine and terbinafine, which are used as antifungal agents, cause similar adverse effects, and this toxicity may limit the potential therapeutic benefit of metabolic disease treatment. This toxicity is attributed to the fact that the site of action of both terbinafine and NB-598 is Y195, and the tertiary amine group in the inhibitor structure forms a hydrogen bond with Y195. This prevents Y195 from interacting with glutamine (Q168) at position 168, inhibiting the conversion of SQLE to the active state. Thus, all the catalytic reactions of SQLE are inhibited, resulting in greater neurological and dermal toxicity (*Padyana et al., 2019*; *Nagaraja et al., 2020*). The IC50 values of allylamine inhibitors in mammalian cells are several orders of magnitude greater than those in fungi, and large doses are often required to achieve therapeutic efficacy, such as antitumor effects (*Ryder, 1988*). Hence, there is a need for careful assessments of the tolerance to adverse effects. NB-598 is a preclinical drug without much safety or pharmacological data. FR194738 is derived from NB-598 and has an IC50 of only 2.1 nM, making it one of the most potent inhibitors. Compared with NB-598, FR194738 has a similar potency to NB-598 in terms of its ability to inhibit cholesterol synthesis, but it has improved lipophilic and pharmacokinetic properties (*Sawada, Washizuka & Okumura, 2004*; *Zhang et al., 2024*). As a specific SQLE inhibitor, FR194738 has little effect on cholesterol upstream or downstream (*Sawada et al., 2001*). The SQLE-specific inhibitor FR194738 attenuates the growth of desmoplasia-resistant prostate cancer *in vitro* in PC3 cells and *in vivo* in a mouse xenograft model harboring PC3 cells (*Shangguan et al., 2022*). Like its parent compound, FR194738 is a preclinical drug with limited safety and pharmacologic data that require additional basic and clinical studies.

## Natural compounds and their derivatives

Many natural compounds and their derivatives may be clinically safe SQLE inhibitors that can effectively and selectively inhibit SQLE activity. For example, *Abe et al. (2000)* reported that green tea polyphenols, the main component of which is galloyl-containing epigallocatechin gallate (EGCG), noncompetitively inhibit SQLE by scavenging reactive oxygen species from the active site of the enzyme. The team synthesized galloyl groups, such as dodecyl ester and gallate dodecyl ester, as SQLE inhibitors, which are widely used in food additives for antioxidant purposes. The metabolites of EGCG are also inhibitory,

and other plant extracts, such as beta-carotene, anthocyanins, tannins, fo-ti, and rhubarb, are also rich in galloyl groups. Grape skins and red wine are rich in the galloyl polyphenolic compound resveratrol, which reversibly and noncompetitively inhibits SQLE activity, with cholesterol-lowering and cardiovascular disease-preventive effects. Although EGCG still has few side effects when consumed at high doses, it has low bioavailability and a short half-life to reach effective therapeutic concentrations.

Ellagitannin analogs of pedunculin and eugenol also showed significant inhibitory efficacy. *Gupta & Porter (2001)* reported that garlic and five of its compounds—S-allylcysteine, alliin, diallyl disulfide, selenocystine, and diallyl trisulfide—were effective in inhibiting SQLE. Unlike the mechanism of inhibition by tea polyphenols, the inhibitory effect of garlic extract on SQLE is irreversible. This is because the arylcysteine in garlic binds to the active site of the SQLE, leading to its inactivation. Additionally, telluride, which is present at high levels in garlic, is not highly selective for SQLE. Instead, it can interact with other proteins and inhibit SQLE activity in Schwann cells *via* the blood–brain barrier. This interaction leads to the inhibition of cholesterol synthesis and the accumulation of squalene, which can adversely affect myelin sheath formation and severely impede neurotransmission. This ultimately results in peripheral ganglionic neural degeneration, segmental demyelination, and paralysis of peripheral nerves (*Wagner, Toews & Morell, 1995*; *Laden & Porter, 2001*). Moreover, different types of SQLE inhibitors offer diverse frameworks for creating novel compounds that mitigate side effects and enhance affinity. However, further investigations are needed to determine their therapeutic potential in treating metabolic diseases.

## CONCLUSIONS

This study comprehensively explored the important role of SQLE in metabolism-related diseases, especially its function in several biological processes, such as cholesterol synthesis, the regulation of the gut microbiota, and tumor immunity. SQLE is not only widely expressed in various tissues and organs but also participates in the onset and progression of numerous metabolism-related diseases, such as NAFLD, cancer, DM, and obesity. These findings emphasize the importance of SQLE as a potential therapeutic target, especially in targeted therapies for these chronic diseases. Inhibitors of SQLE (*e.g.*, terbinafine, NB598, and several natural compounds and their derivatives) show promising therapeutic potential, as they are effective in improving SQLE-related pathologies and offer novel strategies for treating metabolism-related diseases. Therefore, an in-depth study of the mechanism of action of SQLE can aid in the development of more targeted drugs to enhance its therapeutic effect.

However, there are several limitations in this study. First, the precise mechanism of SQLE in different metabolism-related diseases remains unclear. In particular, the regulation of SQLE expression in different tissues and its effects on metabolic pathways have yet to be explored in detail. In addition, the clinical application of SQLE inhibitors is still in its infancy, and more randomized controlled trials are needed to verify the safety and efficacy of long-term use. Future research should focus more on these aspects, especially the development of large-scale clinical trials, to provide a more solid scientific

basis for SQLE-targeted therapy. Future research directions could further explore the biomarker potential of SQLE in metabolism-related diseases, especially in early diagnosis and prognostic assessment. In addition, attention should focus on the interactions between SQLE and other metabolic pathways, such as its relationship with the intestinal flora and its regulatory mechanisms under different pathological conditions, which will help construct a more comprehensive metabolic disease model and provide theoretical support for the development of novel therapies.

In summary, the study of SQLE not only provides new perspectives for understanding the mechanism of metabolic-related diseases but also paves the way for the development of targeted drugs. With further elucidation of the biological functions of SQLE and an in-depth study of its relationship with metabolic-related diseases, it is expected that more effective therapeutic strategies will be identified to improve the quality of life and health of patients in the future. To conclude, targeted therapy with SQLE provides an important scientific basis for the prevention and treatment of metabolic-related diseases and has broad application prospects and practical significance.

## ACKNOWLEDGEMENTS

Thank you to every member of the lab.

### Funding

This work was supported by the National Natural Science Foundation of China (No. 81973585). The funders had no role in study design, data collection and analysis, decision to publish, or preparation of the manuscript.

### Grant Disclosures

The following grant information was disclosed by the authors:
National Natural Science Foundation of China: 81973585.

### Competing Interests

The authors declare that they have no competing interests.

### Author Contributions

- Mingzhu Chen conceived and designed the experiments, performed the experiments, analyzed the data, prepared figures and/or tables, authored or reviewed drafts of the article, and approved the final draft.
- Yongqi Yang performed the experiments, analyzed the data, prepared figures and/or tables, authored or reviewed drafts of the article, and approved the final draft.
- Shiting Chen analyzed the data, authored or reviewed drafts of the article, and approved the final draft.
- Zhigang He analyzed the data, authored or reviewed drafts of the article, and approved the final draft.

- Lian Du conceived and designed the experiments, performed the experiments, authored or reviewed drafts of the article, and approved the final draft.

## Data Availability

This is a literature review.

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
