# Peer review of "Targeting squalene epoxidase in the treatment of metabolic-related diseases: current research and future directions"

_PeerJ, doi:10.7717/peerj.18522_

## Round 0.1 · original submission · Major Revisions

The paper provides a comprehensive overview of the recent research progress on the role of Squalene Epoxidase (SQLE) in metabolic-related diseases. I have some suggestions:

1. The paper mentions several SQLE inhibitors, but its discussion of novel or emerging inhibitors is incomplete. Recent advancements in drug discovery and design could offer new avenues for targeting SQLE, potentially overcoming limitations of existing inhibitors. A more thorough examination of these emerging compounds would enrich the review.
2. While the paper provides an extensive review of the preclinical research on SQLE and its role in metabolic diseases, it lacks comprehensive clinical data supporting the therapeutic potential of SQLE inhibitors in humans.

Reviewer 1 ·

Basic reporting

The article "Research progress in the targeted treatment of metabolic-related diseases with SQLE" is an review article describing the importance of SQLE in cholesterol metabolism which is significantly involved in pathogenesis in numerous diseases.
The article title could be improved such as
"Targeting SQLE in the treatment of metabolic-related diseases: current research and future directions",
" Innovations in metabolic disease management - the therapeutic potential of SQLE inhibition" or something like that.
The article is well structured, figures and table are well presented.
Authors could improve the section of NAFLD by adding the potential role of SQLE inhibition on improving insulin resistance if there are available research.
So far, there are numerous papers regarding SQLE role in carcinomas, so this article can be more focused on metabolic disorders.

Experimental design

Study is well designed in order to evaluate the recent literature to summarize the potential role of SQLE in obesity, diabetes, NAFLD and cancers. Methodology is also described well, included all databases and adequate keywords for literature research.
All included paragraphs are sufficient, and well described.
However, the authors should reorganize some sections. It would be better to merge paragraphs describing the role of SQLE in obesity, DM, NAFLD and cancer with same paragraphs with clinical significance.
Additionally, authors should include some clinical trials, or more clinical studies that are considering SQLE inhibitors in patients if possible. Moreover, it would be nice to add one more table comparing basic and clinical studies and effects of SQLE inhibitors.

Validity of the findings

The main finding of the study is actually the huge potential of SQLE as a therapeutic target in various diseases, especially malignancy. Authors should provide a list or a table with so far known therapeutic agents and its effects.
Furthermore, authors could add a paragraph regarding Future directions.

Additional comments

The article has good potential and is evaluating very important topic. With major corrections the article can be published.

·

Basic reporting

1. The manuscript title should not contain abbreviations without providing the full terms.
2. Please add references for the paragraphs in lines 63-67 and 79-82.
3. Provide full terms for SCAP and COPII in lines 135 and 137.
4. Ensure that the full term for every abbreviation is provided at its first appearance. Please thoroughly check the entire manuscript.
5. Replace “sterol regulatory element (SRE)” with “SRE” in line 40.
6. Clarify whether Figure 2 is the authors' original work. If so, add a note or provide a reference to avoid plagiarism.
7. The details in Figure 2 do not align with the content under Transcriptional Regulation. Clarify what MYC is, and address the confusing feedback mechanism.
8. Include legends and captions for Figure 2.
9. Add references for the paragraphs in lines 214-218, 323-327, 431-432, 432-435, and 484-493.
10. Clarify whether SREBPs and SREBP2 are the same.
11. In Figure 1, many enzymes are involved in cholesterol synthesis. Explain why only HMGCR and SQLE are considered important (line 109).
12. Adjust the placement of “Lanosterol synthesis” in Figure 1 to avoid overlapping with the right-side content. Link the final product of the “Shunt pathway” to “cholesterol” to show the importance of HMGCR and SQLE.
13. Confirm if Dioxidosqualene in Figure 1 is the same as deoxygenated squalene (line 112).
14. In Table 1, miRNA-205 does not match miR-205 in the manuscript. The reference format also does not match, making it difficult to trace the sources.
15. Clarify the meaning of the heading “miRNA expression level SQLE expression” in Table 1.
16. Table 1 lists mRNA-584-5p and mRNA363-3p, but they are not mentioned in the manuscript. Please clarify.
17. Lines 151 and 155 discuss Qian et al.'s study on miR-133b in pancreatic and esophageal cancer, as shown in Table 1. However, lines 155 mention SQLE in breast cancer (Qin et al.), which is not reflected in Table 1.
18. Follow line 162 with (Figure 2). circ_0000182 is not shown in Table 1.
19. The content under Posttranslational Regulation (lines 169-209) does not align well with Figure 2. Revise Figure 2 to match the manuscript content.
20. The statement in line 503 about allicin upregulating SQLE and demonstrating a therapeutic effect on hyperlipidemia seems contradictory. If allicin upregulates SQLE, it should increase cholesterol synthesis, not reduce it.
21. The statements in lines 512-516 about chlorophyll-like substances causing hepatorenal toxicity yet being considered therapeutic agents seem contradictory. Please clarify.
22. In lines 518-519, the claim that “SQLE is expected to be a highly selective drug target with few side effects” requires supporting evidence. Please provide it.

Experimental design

No comment.

Validity of the findings

No comment.

Additional comments

The subject of this study is interesting and has promising potential. However, the content is complex, involving numerous genes and enzymes, which may confuse readers. Incorporating well-designed figures that align closely with the text will help make the material easier to follow and understand.

---

## Round 0.2 · Minor Revisions

Authors should address some issues raised by reviewer2.

Reviewer 1 ·

Basic reporting

the authors have improved manuscript as suggested. In this improved form, it can be accepted for publication.

Experimental design

/

Validity of the findings

/

·

Basic reporting

Reviewer’s Comments
1. The Abstract should not contain references. Please remove all references from the Abstract.
2. The caption for Figure 2 states: “The figure is adapted from Zou et al. (2024) with modifications. The figure is adapted from Zou et al. (2022) with modifications (Zou et al. 2022).” However, Zou et al. (2024) is not listed in the References. Please clarify.
3. Lines 67-68 need adjustment.
4. Lines 128-137 contain the name and caption of Figure 1, which are already presented with the figure. Please delete these lines.
5. Lines 205-212 contain the name and caption of Table 1, which are already presented with the table. Please delete these lines.
6. Lines 258-264 contain the name and caption of Figure 2, which are already presented with the figure. Please delete these lines.
7. Lines 587-591 contain the name and caption of Table 2, which are already presented with the table. Please delete these lines.
8. The sections ‘The role of the SQLE in metabolic-related diseases’ (lines 272-401) and ‘Clinical correlation between the SQLE and metabolic diseases’ (lines 430-498) are repetitive and overlap. I suggest combining these sections, with each sub-heading starting with basic knowledge followed by clinical correlation and implications. This will make the manuscript more concise and easier to understand.
9. Please rewrite the Conclusion to make it concise. The Conclusion should not repeat the Discussion or include references. The conclusion of a review article typically includes several key elements: 9.1. Summary of Key Findings: A concise recap of the main points and findings discussed in the review. 9.2. Implications: Discussion on the broader implications of these findings for the field, including practical applications and theoretical contributions. 9.3. Limitations: Acknowledgment of any limitations in the reviewed studies or the review process itself. 9.4. Future Research Directions: Suggestions for areas where further research is needed to address gaps or unresolved questions. 9.5. Final Thoughts: A concluding statement that encapsulates the overall significance of the review and its contributions to the field.

Experimental design

No comment.

Validity of the findings

No comment.

Additional comments

The manuscript needs to be rewritten to enhance conciseness and clarity.

---

## Round 0.3 · Minor Revisions

The revised manuscript be reviewed by all co-authors before resubmission. All minor errors should be addressed.

·

Basic reporting

Reviewer’s Comments on Revision 1
1. The authors have revised the Conclusions according to my previous comments. However, several significant mistakes remain:
1.1. The contents in lines 566-574 are unrelated to the manuscript's topic of metabolic-related diseases and do not appear elsewhere in the manuscript.
1.2. References still appear inappropriately.
2. Lines 129-130 should read: "There are three SREBPs:"
3. I suggest the revised manuscript be reviewed and approved by all co-authors before resubmission to the Editorial.

Experimental design

No comment.

Validity of the findings

No comment.

Additional comments

No comment.

---

## Round 0.4 · accepted · Accept

Congratulations! This paper is ready for publication.

·

Basic reporting

All comments were appropriately addressed.

Experimental design

No comment.

Validity of the findings

No comment.